# 1 Chamber simulation on the formation of secondary organic

## 2 aerosols (SOA) from diesel vehicle exhaust in China

- Wei Deng<sup>1,2</sup>, Qihou Hu<sup>1</sup>, Tengyu Liu<sup>1,2</sup>, Xinming Wang<sup>1,\*</sup>, Yanli Zhang<sup>1</sup>, Xiang Ding<sup>1</sup>, Yele Sun<sup>3</sup>,
- Xinhui Bi<sup>1</sup>, Jianzhen Yu<sup>4</sup>, Weiqiang Yang<sup>1,2</sup>, Xinyu Huang<sup>1,2</sup>, Zhou Zhang<sup>1,2</sup>, Zhonghui Huang<sup>1,2</sup>,
- Quanfu He<sup>1,2</sup>, A. Mellouki<sup>5</sup>, Christian George<sup>6</sup>
- <sup>1</sup>State Key Laboratory of Organic Geochemistry and Guangdong Key Laboratory of Environmental
- Protection and Resources Utilization, Guangzhou Institute of Geochemistry, Chinese Academy of
- Sciences, Guangzhou 510640, China
- <sup>2</sup>University of Chinese Academy of Sciences, Beijing 100049, China
- <sup>3</sup>Institute of Atmospheric Physics, Chinese Academy of Sciences, Beijing 100029, China
- <sup>4</sup>Division of Environment, Hong Kong University of Science & Technology, Clear Water Bay,
- Kowloon, Hong Kong, China
- <sup>5</sup>Institut de Combustion, A érothermique, R éactivit é et Environnement (ICARE), CNRS, 45071 Orl éans
- cedex 02, France
- <sup>6</sup>Institut de Recherches sur la Catalyse et l'Environnement de Lyon (IRCELYON), CNRS, UMR5256,
- Villeurbanne F-69626, France
- \*Corresponding author:
- Dr. Xinming Wang
- State Key Laboratory of Organic Geochemistry
- Guangzhou Institute of Geochemistry, Chinese Academy of Sciences
- Tel: +86-20-85290180; Fax: +86-20-85290706
- Email: wangxm@gig.ac.cn

# 25 Abstract

| 26 | In China primary particulate matter emission from on-road vehicles is predominantly               |
|----|---------------------------------------------------------------------------------------------------|
| 27 | coming from diesels, yet secondary organic aerosols (SOA) formed from diesel                      |
| 28 | emission may be also of greater significance due to more intermediate volatile organic            |
| 29 | compounds (IVOC) in the exhaust. Here we introduced exhaust from in-use diesel                    |
| 30 | vehicles under warm idling condition directly into an indoor smog chamber with a                  |
| 31 | 30m <sup>3</sup> Teflon reactor, and investigated the SOA formation as well as chemical aging of  |
| 32 | organic aerosols during photo-oxidation. The emission factors of primary organic                  |
| 33 | aerosol (POA) and black carbon (BC) for the three typical Chinese diesel vehicles                 |
| 34 | ranged 0.18-0.91 and 0.15-0.51 g kg-fuel <sup>-1</sup> , respectively; and the SOA production     |
| 35 | factors ranged 0.50-1.8 g kg-fuel <sup>-1</sup> with an average SOA/POA ratio of 1.6. Aromatic    |
| 36 | hydrocarbons could only explain less than 3% of SOA formed during aging, and                      |
| 37 | IVOC and oxygenated VOC might contribute substantially to SOA formation. High                     |
| 38 | resolution time-of-flight aerosol mass spectrometer (HR-ToF-AMS) resolved that                    |
| 39 | POA dominated by CH classes (alkanes, cycloalkanes and alkenes) with high                         |
| 40 | abundances of the $C_nH_{2n+1}$ and $C_nH_{2n-1}$ fragments, and after photo-oxidation the        |
| 41 | fraction of CH classes and the H/C ratios decreased, while the fraction of CHO, as                |
| 42 | well as the ratios of O/C and of organic matter to organic carbon (OM/OC), all                    |
| 43 | increased. The plot of $f_{44}$ (ratio of $m/z$ 44 to the total signal in a mass spectrum) versus |
| 44 | $f_{43}$ indicated that diesel SOA were semi-volatile oxygenated organic aerosols                 |
| 45 | (SV-OOA). The slopes of O:C versus H:C element ratios in the Van Krevelen diagram                 |
| 46 | ranged from -0.47 to -0.68, suggesting a combination of carboxylic acid and                       |

- alcohols/peroxides formed during the aging of diesel exhaust.

# 49 **1 Introduction**

| 50 | Air pollution by particulate matter not only adversely affects human health by causing      |
|----|---------------------------------------------------------------------------------------------|
| 51 | respiratory and cardiopulmonary diseases (Pope et al., 2009; Brook et al., 2010; Liu et     |
| 52 | al., 2015b; Lelieveld et al., 2015), but also impacts regional and global climate           |
| 53 | (Ramanathan et al., 2001; Parrish and Zhu, 2009; Wang et al., 2014b). Health risks are      |
| 54 | of particular concern when heavy fine particle (particulate matter with dynamic             |
| 55 | diameter less than 2.5 $\mu m,PM_{2.5})$ pollution occurs in densely populated megacities,  |
| 56 | such as China's capital city Beijing, which is hard-hit by frequent heavy haze episodes     |
| 57 | with a large body of people exposed to severe $PM_{2.5}$ pollution (Guo et al., 2014;       |
| 58 | Huang et al., 2014). In urban agglomerations, vehicle exhaust contributes                   |
| 59 | substantially to $PM_{2.5},$ with mass fractions ranging from ~22% in southeastern US       |
| 60 | (Chen et al., 2012), ~37% in Guangzhou in the Pearl River Delta during wet season           |
| 61 | (Cui et al., 2015), to as high as 49% in Mexico City (Stone et al., 2008). In particular,   |
| 62 | People usually expose to much higher air pollutants in urban roadside                       |
| 63 | microenvironments due to traffic-related emission (Zhao et al., 2004; Xu et al., 2008).     |
| 64 | Nevertheless, the contribution of vehicle exhaust to $PM_{2.5}$ is often a debatable issue. |
| 65 | In Beijing, for example, previous studies revealed that contributions of vehicle            |
| 66 | exhaust to $PM_{2.5}$ might range from 4% to 16.3% (Zheng et al., 2005; Song et al.,        |
| 67 | 2006a, b, 2007b; Zhang et al., 2013; Wu et al., 2014), whilst very recently Beijing         |
| 68 | Municipal Environmental Protection Bureau announced that vehicle exhaust alone              |
| 69 | accounted for 31% of PM <sub>2.5</sub> mass (http://www.bjepb.gov.cn/bjepb/413526/331443/   |
| 70 | 331937/333896/396191/index.html). One crucial reason for the discrepancies is the           |

lack of understanding about secondary aerosols formed from vehicle exhaust.

Direct motor vehicle emission of PM is predominantly from diesel vehicles (Reff et al., 2009; Zhang et al., 2009). In China diesel vehicles contributed more than 99% of 73 primary vehicle emission of PM although they only account for 15.2% of China's 74 75 on-road vehicles (MEPC, 2014). Recent studies in Beijing revealed that diesel vehicles contribute 80%-90% of PM emissions from on-road sources (Huo et al., 76 77 2011; Wu et al., 2010; Wang et al., 2010). Hence, restriction of diesel vehicles into the 78 core urban areas has become a control measure widely adopted by municipal 79 governments to improve air quality. Besides primary particle emission, vehicle exhaust also contributes substantially to gaseous pollutants, such as volatile organic 80 compounds (VOCs) and nitrogen oxides (NOx), which can form secondary organic 81 and inorganic aerosols via photo-oxidation (Weitkamp et al., 2007; Robinson et al., 82 2007; Nordin et al., 2013; Liu et al., 2015a). Nordin et al. (2013) reported that 83 secondary organic aerosols (SOA) formed from gasoline exhaust can reach as high as 84 500 times that of primary organic aerosols (POA). Although primary PM emission 85 86 factors of diesel vehicles are typically orders of magnitude higher than gasoline vehicles, recent studies demonstrated that for diesel vehicles the SOA/POA ratios 87 could reach about 3 based on chamber simulations (Chirico et al., 2010; Gordon et al., 88 2014b). Consequently, contribution of vehicle exhaust to ambient fine particles would 89 90 become more complicated if considering secondary aerosol formation.

In China, a large portion of gasoline vehicles are produced in Sino-Foreign joint
ventures and due to transfer of gasoline engine technology from abroad, chamber

simulation study showed that the SOA/POA ratios for China's gasoline vehicle 93 94 exhaust are quite similar with those reported in the Europe or in the US (Liu et al., 2015a). However, engines equipped on China's diesel vehicles are mainly designed 95 and produced domestically with their technology lagging behind the developed 96 97 nations. According to previous studies (Yanowitz, 2000; Cheung et al., 2009; Liu et al., 2009), the emission factors of both hydrocarbon and particulate matter for diesel 98 99 vehicles in China were much higher than those in the developed nations. Therefore, 100 the SOA formation from China's diesel exhaust may be different with those in Europe 101 and the US as well. Furthermore, most diesel vehicles in China are not equipped with emission control aftertreatment devices, which can significantly reduce both POA 102 emission and SOA formation (Chirico et al., 2010; Gordon et al., 2014b). As previous 103 study indicates that even for diesel vehicles SOA might dominate over POA, 104 formation of SOA from diesel vehicles in China would be an issue of wide concern. 105 In this study, we chose three typical types of diesel vehicles made in China, 106 introduced the exhaust from the diesel vehicles under warm idling condition into an 107 indoor smog chamber with a 30 m<sup>3</sup> reactor, and investigated the SOA formation under 108

photo-oxidation. The main purpose of this study is to obtain a more comprehensive
evaluation of diesel vehicle's contribution to carbonaceous aerosols by studying SOA
formation from the primarily emitted exhaust.

## 112 2 Materials and methods

#### 113 2.1 Vehicles and fuel

Table 1 lists the three diesel vehicles used for our chamber experiments. They

represent three different types of diesel vehicles manufactured by three major diesel 115 116 vehicle makers in China. Foton is a medium-duty passenger vehicle made by the Baic Motor Corporation LTD., Changan is a medium-duty truck made by the China 117 Changan Automobile Group, and JAC is a heavy-duty truck made by the JAC Motors. 118 119 In 2011, the diesel vehicle productions of the three companies were 490,280, 299,506 and 230,452, respectively (China Automotive Industry Yearbook, 2012), and their 120 121 diesel vehicle sales were all among the top 10 in China. All the vehicles in this study 122 had no exhaust aftertreatment devises and they were fueled with Grade 0# diesel, 123 which complies with the Euro III diesel fuel standard.

### 124 2.2 Experimental setup

The experiments were carried out in the indoor smog chamber at Guangzhou Institute 125 of Geochemistry, Chinese Academy of Sciences (GIG-CAS) with a ~30 m<sup>3</sup> Teflon 126 reactor suspended in a temperature-controlled room. Details of setup and facilities 127 about the chamber were described elsewhere (Wang et al., 2014a). Briefly, 135 black 128 lamps (1.2 m long, 60 W Philips/ 10R BL, Royal Dutch Philips Electronics Ltd., the 129 Netherlands) are used as light source, providing a NO<sub>2</sub> photolysis rate of 0-0.49 min<sup>-1</sup>. 130 Temperature can be set in a range from -10 to 40°C with an accuracy of  $\pm 1$ °C, and is 131 measured by eight sensors inside the enclosure and the other one inside the Teflon 132 reactor. In this study, temperature and relative humidity (RH) for all experiment were 133 134 set to 25°C and less than 5%, respectively. Prior to each experiment, the Teflon chamber was flushed with dry purified air for at least 48 hours, which represents at 135 least 5 whole exchanges of the reactor volume. Before each experiment, the chamber 136

- was checked for hydrocarbons, ozone,  $NO_x$ , and particles inside the reactor to make
- sure it was clean.

Before introducing exhaust into the chamber reactor, all vehicles in the experiments were at "warm idling" mode, which means the vehicles were started and run on-road for about 30min before staying at idling condition. Depending on the organic aerosol concentration and particle numbers reached inside the reactor as sensed by the affiliated instruments, the exhaust injection time ranged from 5 to 20 min.

After introducing exhausts, nitrous acid (HONO) was bubbled into the chamber as a source of hydroxyl radical (OH). Propene was added to adjust the VOC/NOx ratios to 145 approximately 3:1 ppbC:ppb. Propene has often been added to adjust VOC/NO<sub>x</sub> ratio 146 in diesel exhaust chamber experiments (Chirico et al., 2010; Presto et al., 2014; 147 Gordon et al., 2014b) and is not considered to be a relevant SOA precursor (Odum et 148 al., 1996; Coker et al., 2001). 60 ppbv of deuterated butanol (butanol-d9) was also 149 injected into the chamber as an OH tracer by using  $k_{\text{butanol-d9}} = 3.4 \times 10^{-12} \text{ cm}^3$ 150 molecule<sup>-1</sup>s<sup>-1</sup> (Barmet et al., 2012; Gordon et al., 2014a). After characterizing the 151 152 primary emissions in a dark condition for an hour, the exhaust was photo-oxidized for 5 h by being exposed to black lights. 153

## 154 2.3 Instrumentation

An array of instruments was used to monitoring trace gases and particles inside the chamber. Ozone (O<sub>3</sub>) was measured with an ozone analyzer (EC9810, Ecotech, Australia) and NO<sub>x</sub> were measured with a trace nitrogen oxides analyzer (EC9841, Ecotech, Australia). SO<sub>2</sub> was measured with a dedicated analyzer (Model 43i, Thermo

Scientific. USA). VOCs were 159 measured online with commercial а 160 proton-transfer-reaction time-of-flight mass spectrometer (PTR-TOF-MS, Model 2000, Ionicon Analytik GmbH, Austria) (Lindinger et al., 1998; Jordan et al., 2009). 161 Offline VOC samples were also collected using 2L stainless steel canisters each 30 162 min during the photo-oxidation, and measured by a Model 7100 Preconcentrator 163 (Entech Instruments Inc., California, USA) coupled with an Agilent 5973N gas 164 165 chromatography-mass selective detector/flame ionization detector (GC-MSD/FID, 166 Agilent Technologies, USA). CO in the canister samples was analyzed using a gas 167 chromatography (6980GC, Agilent, USA) with a flame ionization detector and a packed column (5A molecular sieve 60/80 mesh, 3 m×1/8 inch). Detailed procedures 168 for the offline analysis of VOCs and CO were described elsewhere (Zhang et al., 169 170 2012). Before and after introducing exhaust, air samples were collected into 3L 171 cleaned Teflon bags to determine CO<sub>2</sub> concentrations with an HP 4890D gas chromatography (Yi et al., 2007). 172

A scanning mobility particle sizer (SMPS, Model 3080 classifier, model 3775 CPC; 173 174 TSI Inc., Minnesota, USA) was used to measure particle number and volume concentrations and size distributions. The particle mass concentration was estimated 175 assuming spherical particles and a density of 1.0 g cm<sup>-3</sup> (Weitkamp et al., 2007). BC 176 concentrations were measured with a seven-channel Aethalometer (Model AE-31, 177 178 Magee Scientific, Berkeley, California). The Aethalometer data were corrected for particle loading effects using the method of Kirchstetter and Novakov (2007). A 179 high-resolution time-of-flight aerosol mass spectrometer (HR-ToF-MS, Aerodyne 180

| 181 | Research Inc., USA) operated in alternating mode were used to measure nonrefractory             |
|-----|-------------------------------------------------------------------------------------------------|
| 182 | submicron aerosol mass and chemical compositions (Jayne et al., 2000; DeCarlo et al.,           |
| 183 | 2006). The average operating time was 1 min for the high-sensitivity V mode and 1               |
| 184 | min for high-resolution W mode. The toolkit Squirrel 1.53G was used to analyze time             |
| 185 | series of various mass components, and Pika 1.12G was used to determine the average             |
| 186 | element ratios ( <u>http://cires1.colorado.edu/jimenez-group/ToFAMSResources/</u>               |
| 187 | ToFSoftware/index.html). For elemental analysis, the data were analyzed based on the            |
| 188 | method described in Aiken et al. (2007, 2008). The fragmentation table from Allan et            |
| 189 | al. (2004) was used to interpret the AMS data. The contribution of gas phase $\mathrm{CO}_2$ to |
| 190 | the AMS $m/z$ 44 signal was corrected by analyzing HEPA filtered air from the smog              |
| 191 | chamber after filling the exhaust.                                                              |

## 192 2.4 Operation Steps

Each experiment consisted of five steps: 1) Introducing exhaust into the chamber from 193 t=-2h. With the injection of exhausts, concentrations of NO<sub>x</sub>, BC and OA were 194 climbing. Their concentrations when the injection stopped are shown in Table 2. The 195 mixing ratio of  $NO_x$  increased from 0 to ~1 ppmv; the particle number concentration 196 increased fast from ~2 to ~350,000 particles cm<sup>-3</sup>; the total particle mass 197 concentrations increased from ~0 to over 100 µg m<sup>-3</sup>; and the VOC concentrations 198 also slightly increased at this step. 2) Characterizing primary emissions from t = -1.5h. 199 After completion of injection, the increase of NO<sub>x</sub>, BC, OA and VOCs were measured 200 against that of CO<sub>2</sub> and CO, and the emission factors were further calculated based on 201 equation (1). 3) Adding HONO and propene at approximately t = -0.5h, leading to a 202

- moderate increasing of both NO and NO<sub>2</sub>, approximately 300 ppbv for each. 4)
- Turning on the lights at t = 0 h to start the photo-oxidation. Substantial amounts of
- SOA formed at the beginning of this period. 5) Turning off the light at t=5h and
- further characterizing the aged diesel vehicle exhausts in the dark for about 2 hours.
- 2.5 Data analysis
- The emission factors (EF) for various pollutants and the production factors (PF) for
  SOA were calculated on a fuel basis (g kg-fuel<sup>-1</sup>):

210 
$$EF_P \text{ or } PF_P = 10^3 \cdot [\Delta P] / (\frac{[\Delta CO_2]}{MW_{CO_2}} + \frac{[\Delta CO]}{MW_{CO}}) \cdot \frac{C_f}{MW_C}$$
(1)

Where  $[\Delta P]$  is the background corrected pollutant concentration in  $\mu g \text{ m}^{-3}$ ,  $[\Delta CO_2]$ 211 and  $[\Delta CO]$  is the background corrected concentration of CO<sub>2</sub> and CO in the chamber 212 in  $\mu g \text{ m}^{-3}$ .  $MW_{CO2}$ ,  $MW_{CO}$  and  $MW_C$  are the molecular weights of CO<sub>2</sub> (44.1 g mol<sup>-1</sup>), 213 CO (28 g mol<sup>-1</sup>) and carbon (12 g mol<sup>-1</sup>), respectively.  $C_f$  is the carbon intensity of the 214 fuel, which was adopted as 0.87 kg C kg-fuel<sup>-1</sup> for diesel (Chirico et al., 2010). 215 Equation (1) assumes that all carbon in the fuel was converted to  $CO_2$  and CO, and 216 the contribution from VOC was negligible. This assumption was reasonable, because 217 218  $[\Delta CO_2]$  and  $[\Delta CO]$  after introducing exhaust were approximate 100 ppmv and 1 ppmv, respectively, while the increase of VOC was below 5ppbv. The concentrations of 219 hydroxyl radical (OH) during the experiments were inferred from the decay of 220 deuterated butanol measured with the PTR-MS (Atkinson and Arey, 2003). The 221 average OH levels during our experiments were calculated to be approximate 2-5×10<sup>6</sup> 222 molecules cm<sup>-3</sup>, which approached to the levels in the ambient and that in the previous 223 study by Gordon et al. (2014b). 224