# Peer review of "Chamber simulation on the formation of secondary organic"

_Atmospheric Chemistry and Physics, 2016_

## Referee Comment (RC1) · Anonymous Referee #1 · 29 Mar 2016

This article investigates SOA formation from diesel vehicle exhaust, using a photochemical smog chamber. The authors find that SOA production is substantially higher from diesel exhaust than in previous measurements of gasoline vehicle exhaust, indicating a disproportionately high contribution to SOA from diesel vehicles in China. They further conclude that traditional aromatic precursors account for less than 3% of SOA production, and speculate on other possibilities. This is an interesting study, and the investigation of SOA and POA from diesel vehicle exhaust in the context of air quality in Chinese cities will be an important contribution. I would therefore encourage publication, subject to the authors addressing the following comments.

There is no mention at all in the paper as to how (or indeed if) the exhaust was di-

luted with clean air in the chamber, other than the sentence "Depending on the organic aerosol concentration and particle numbers reached inside the reactor as sensed by the affiliated instruments, the exhaust injection time ranged from 5 to 20 min". From this, I understand that the exhaust was injected into the chamber, until the concentrations reached levels that you thought appropriate. Please clarify how the injection time was decided. Please also quote the final dilution ratios, as without these, the numbers given in section 2.4 and table 2, and any comparison between experiments, are meaningless. Do the emission and production factor calculations in sections 2.5 and 3.1 account for dilution? Is the dilution ratio representative of the dilution of diesel exhaust in an urban environment?

Line 145: "Propene was added to adjust the VOC/NOx ratios to approximately 3:1 ppbC:ppb". What concentration of propene was added? Also, I understand that the ratio 3:1 is considered typical of urban environments, but please say so and include references.

Lines 230-232: This needs to be explained better – what is $\omega$?

Section 3.1: Here you state "The relatively backward diesel engine technology and lack of emission aftertreatment devices like diesel oxidation catalyst (DOC) or diesel particulate filter (DPF) would probably be the reasons for higher EFs of POA for China's diesel vehicles in this study", however the EF(POA) reported by the Chirico and Gordon studies were measured without after-treatment (you are aware of this as you state this in the caption of figure 1), so this could not be the reason. Also why do the other studies see higher EF(BC)? In the EF calculation, you assume the same carbon intensity of fuel as Chrico et al. Could this be different?

Line 279: Can the authors speculate as to why the difference here? Higher POA?

Lines 286-290: Have I understood this correctly: there are far more gasoline vehicles than diesel vehicles in China, yet more diesel is consumed for transportation that gasoline?

[Figure]

Line 316: "the discrepancies between predicted and measured SOA were still huge". By how much? Please quantify.

Table 2, and line 718: Please put the units "×106 molecules cm-3" in the column header.

Figure 3: Please include ticks on the y-axis below 100 nm, to make the position of the mode clearer.

[Figure]

---

## Referee Comment (RC2) · Anonymous Referee #2 · 1 Apr 2016

Deng et al present measurements of secondary organic aerosol (SOA) formation from dilute diesel exhaust in a smog chamber. The manuscript is topically relevant to ACP, and the journal has published several similar papers in the past (e.g., Chirico et al 2010, Gordon et al 2014a and b, Nordin et al 2013). However it is not ready for publication in its current form, and needs significant revision.

General comments: My primary criticism is that this manuscript does little to differentiate itself from previous similar works (e.g., Chirico et al 2010, Gordon et al 2014a and b, Nordin et al 2013, and others), and does not seem to offer much in the way of new information or scientific insight. The authors argue that the value of this manuscript lies in the fact that this is the first such study using Chinese diesel engines, which have

fewer pollution controls than modern US and European diesels. I do not find this a compelling argument for publication, as both Gordon et al and Chirico et al conducted nearly identical experiments with diesel vehicles without after treatments such as DOC or DPF.

The authors also argue that the higher SOA production observed from their engines compared to Gordon et al and Chirico et al is a significant result. However, this could merely be a result of less dilution of the exhaust in the smog chamber and resultant partitioning during oxidation. Figure 2 shows that the OA concentration in the chamber after exhaust injection was 50 ug/m^3. This means that the chamber had ample POA for SOA to partition into, and oodles of vapor available to oxidize and form SOA. My impression is that previous similar studies worked with much lower POA concentrations, and therefore produced less SOA. The authors have done nothing to convince me that the excess SOA formed in their experiments is the result of higher SOA formation potential of the exhaust rather than higher initial concentrations.

The results presented in the manuscript show that a significant amount of SOA is formed during photo-oxidation of dilute exhaust, and that the OA becomes more oxidized during oxidation. This is not new news - Sage et al showed this exact behavior in ACP in 2008 (albeit with a laboratory-scale diesel engine), and the result has been repeated multiple times with emissions from many sources (gasoline engines, diesel engines, aircraft, etc).

While I think the underlying data are sound, and the methods appropriate, simply regurgitating previously-published experiments should not pass muster for publication in ACP. The authors need to show how their study adds substantially to the existing state of knowledge.

Specific comments:

- The manuscript needs a thorough English grammar check.

- Is Figure 1 comparing apples to apples? E.g., does it compare the w=0 case from the present study to w=0 from previous work (or w=1 to w=1)?

- Line 330-333: Increase in m/z 59 during oxidation could also be acetone

- Figure 4 - which line goes with which axis?

- The authors only seem to use experiment 8 for representative data. This makes me wary about results from the other experiments.

- Figure 8a - what does SOA mean? Is this the total OA at the end of the experiment or was SOA separated from POA somehow?

- Fig 8b would be better if it showed VK plots for more than one experiment.

---

## Referee Comment (RC3) · Anonymous Referee #3 · 4 Apr 2016

Deng et al. present results showing that diesel exhaust injected into a smog chamber produces SOA when the lights are switched on. This result is far from unexpected, as previous studies have shown the same. Reading through the article gives the impression that the authors followed a recipe. If anything, the main contribution of this work would be to demonstrate that a diesel car without after treatment is the same in China as anywhere else. Thus, considering ACP's scope '[ACP] is focused on studies with general implications for atmospheric science rather than investigations that are primarily of local or technical interest', I feel that this articles falls under the latter category, particularly with regard to local interest.

But perhaps the authors can add something, therefore, aside from criticism of the arti-

cle's scope, there are other major issues of concern:

The grammar needs improvement throughout. While perfection does not need to be the goal, the language should not get in the way of understanding the research.

I also have a concern regarding data represented in Fig. 2C. The authors use the black carbon (BC) time series to quantify organic aerosol (OA) wall losses. Firstly, looking at the first hour after lights on, BC does not decrease. Despite this, wall lost OA is added (since the raw OA trace is decreasing after around 30 minutes). Secondly, variation in the raw trace does not match variation in the wall loss corrected traces. See for example the small perturbation just after t=-1h, not represented in the wall loss corrected trace. Thirdly, variation in the BC trace is not present in the OA trace for the w=1 case. I also note that Equation 2 (w=0 wall loss) does not give the time dependent suspended OA mass, while equation 3 (w=1) does. Can the authors please provide here an equivalent expression for suspended OA mass as a function of time (i.e. the equation used to calculate corrected OA mass), and give assurance that the data, as calculated, is that which is shown?

———————————————

---

## Author Comment (AC1) · 26 Apr 2016

Q: There is no mention at all in the paper as to how (or indeed if) the exhaust was diluted with clean air in the chamber, other than the sentence "Depending on the organic aerosol concentration and particle numbers reached inside the reactor as sensed by the affiliated instruments, the exhaust injection time ranged from 5 to 20 min". From this, I understand that the exhaust was injected into the chamber, until the concentrations reached levels that you thought appropriate. Please clarify how the injection time was decided.

Reply: The exhaust was first diluted by ejector dilutor (DI-1000, Dekati Ltd., Finland), then introduced into the chamber which was pre-filled with purified air. We decided

the injection time by the particle mass in the chamber measured by SMPS. When the particle mass reached about 50 $\mu$g m-3, which is comparable to the annual mean value of PM2.5 in Guangzhou (Guangzhou Environmental Protection Bureau, 2014), we stopped injecting the exhaust. (Lines 140-147 in the revised manuscript).

Q: Please also quote the final dilution ratios, as without these, the numbers given in section 2.4 and table 2, and any comparison between experiments, are meaningless.

Reply: We have added the dilution ratios in the revised Table 2.

Q: Do the emission and production factor calculations in sections 2.5 and 3.1 account for dilution? Is the dilution ratio representative of the dilution of diesel exhaust in an urban environment?

Reply: As the $CO_2$ concentrations used to calculate emission and production factors in equation (1) were background-subtracted, therefore the emission and production factors have taken dilution into account. As shown in Table 2, the dilution ratios ranged from 66-215, which were lower than the dilution of exhaust in an urban environment ($\sim$1000:1 as reported by Zhang et al., 2004). But due to detection limits of instruments, reactants in chamber study are usually higher than those in real-world conditions (Odum et al., 1997; Surratt et al., 2007; Hildebrandt et al., 2009). The dilution ratios were comparable with previous diesel exhaust studies (Chirico et al., 2010; Gordon et al., 2014b).

Q: Line 145: "Propene was added to adjust the VOC/NOx ratios to approximately 3:1 ppbC:ppb". What concentration of propene was added? Also, I understand that the ratio 3:1 is considered typical of urban environments, but please say so and include references.

Reply: The concentrations of introduced propene were added to Table 1. The VOC/NOx ratios of approximately 3:1 ppbC:ppb is considered as an typical ratio for urban environments (Guo et al., 2013). We have added the reference in the revised

manuscript.

Q: Lines 230-232: This needs to be explained better – what is w? Reply: $\omega$ is a proportionality factor of organic vapor partition to chamber walls and suspended particles. We have explained it in the revised manuscript.

Q: Section 3.1: Here you state "The relatively backward diesel engine technology and lack of emission aftertreatment devices like diesel oxidation catalyst (DOC) or diesel particulate filter (DPF) would probably be the reasons for higher EFs of POA for China's diesel vehicles in this study", however the EF(POA) reported by the Chirico and Gordon studies were measured without after-treatment (you are aware of this as you state this in the caption of figure 1), so this could not be the reason. Also why do the other studies see higher EF(BC)? In the EF calculation, you assume the same carbon intensity of fuel as Chrico et al. Could this be different?

Reply: Yes, the lack of emission aftertreatment devices could not be the reason for the difference of EFs(POA) between this study and previous studies. As the diesel fuel used in this study is at Euro III standard, in which the sulfur content is approximately 350ppm (Zhang et al., 2010; Yue et al., 2015), which is much more higher than that in previous studies (Euro V). However, the fuel sulfur content can affect the particle emission for diesel vehicle (Rönkkö et al., 2007). Therefore, the difference in engine technology and fuel quality could be the reason of higher emission factors of POA. When at idling condition for the vehicle without aftertreatment, the EF(BC) of Changan were 0.47-0.51 g kg-fuel-1, which were similar to Chirico et al. (2010) (0.466-0.763 g kg-fuel-1), the EF(BC) of Foton and JAC (0.15-0.19 g kg-fuel-1) were comparable to Gordon et al. (2014b) ($\sim$0.260 g kg-fuel-1). Moreover, Zheng et al. (2015) reported EF(BC) for Euro III HDDV as 612$\pm$740 mg kg-fuel-1. The EF(BC) could be varied vehicle by vehicle. Generally, we can conclude that EF(BC) in this paper was comparable to the results from previous studies. The carbon intensity is the mass fraction of carbon in the diesel fuel; it is related to the chemical composition of diesel. As the diesel is mainly comprised of C10-C22 alkane, alkene, aromatics and PAHs, we can calculate the carbon intensity of alkane (0.848 $\pm$ 0.003), alkene (0.857), aromatic (0.885 $\pm$ 0.011) and PAH (0.943 $\pm$ 0.005). Applied the diesel composition in China reported by Yue et al. (2015) (alkene, 20.5%; aromatic, 29.6%; PAH, 8.8%; remained fraction were treated as alkane, 41.1%), the calculated carbon intensity is 0.869, near to the value of fuel as Chrico et al. and this study.

Q: Line 279: Can the authors speculate as to why the difference here? Higher POA?

Reply: As the experiments conditions (VOC/NOx ratios, OH concentrations) were similar to previous studies, and the photochemistry was also the same according to the manuscript, therefore, the reason of difference could be the higher POA. In addition, the O:C ratios of POA in this study (0.3-0.5) were higher than previous study ($\sim$0.2) (Chirico et al., 2010). As the volatility of organics and O:C are generally inversely correlated (Lanz et al., 2007; Jimenez et al., 2009; Ulbrich et al., 2009), this demonstrated that less organics would partition to gas phase and more organics would partition to particle phase (i.e., POA) during the introduction and dilution of exhaust (Donahue et al., 2006). Moreover, the reaction rate constant in gas phase is higher than that in particle phase (Esteve et al., 2006; Bedjanian et al., 2010). Therefore, the difference in the compositions of primarily emitted organics could be the reason.

Q: Lines 286-290: Have I understood this correctly: there are far more gasoline vehicles than diesel vehicles in China, yet more diesel is consumed for transportation that gasoline?

Reply: Yes. There are far more gasoline vehicles than diesel vehicles in China. While the gasoline-consuming transportation only contains road transport (Hao et al., 2015), the diesel-consuming transportation includes not only road transport, but also railway transport and waterway transport. Moreover, the emission/production factors of POA/SOA for train and ship were much higher than that for diesel vehicle (Hallquist et al., 2013; Krasowsky et al., 2015). The information of fuel consumption for motor vehicle was quite limited in China. According to Ou et al. (2010), the fuel consumption

of gasoline and diesel in road transport sector were 52.20 and 38.53 million tons in 2007. With these figures the gasoline and diesel derived OA are estimated to be 12.32 and 38.53 thousand tons, respectively.

Q: Line 316: "the discrepancies between predicted and measured SOA were still huge". By how much? Please quantify.

Reply: If the SOA yield was assumed to be 30% (Gordon et al., 2014b), the ratio of predicted SOA to measured SOA ranged from 1.6% to 8.9%. The following text has been added to revised manuscript. "aromatics only accounted for less than 10% of total SOA."

Q: Table 2, and line 718: Please put the units "106 molecules cm-3" in the column header.

Reply: Revised as suggested (Table 2).

Q: Figure 3: Please include ticks on the y-axis below 100 nm, to make the position of the mode clearer.

Reply: Revised as suggested.

---

## Author Comment (AC2) · 26 Apr 2016

Q: My primary criticism is that this manuscript does little to differentiate itself from previous similar works (e.g., Chirico et al 2010, Gordon et al 2014a andb, Nordin et al 2013, and others), and does not seem to offer much in the way of new information or scientific insight. The authors argue that the value of this manuscript lies in the fact that this is the first such study using Chinese diesel engines, which have fewer pollution controls than modern US and European diesels. I do not find this a compelling argument for publication, as both Gordon et al and Chirico et al conducted nearly identical experiments with diesel vehicles without after treatments such as DOC or DPF.

Reply: Thanks for the comments. China is facing a dilemma in combating air pollu-

tants from traffic emission. We have much more gasoline vehicles than diesel ones. This would be good to reduce PM emission from motor vehicles. However, in recent years surface ozone pollution is becoming more and more serious in many Chinese megacities. Like the situation in Europe, replacing more gasoline cars with diesel ones in China would certainly benefit ground ozone control. We do not know how much SOA would be formed from diesel exhaust. We must pay attention to this point since OA is already the most abundant component in PM2.5 in most of China cities. As diesel engines in China are mostly homemade, we should take cautions if we use results based on diesels in USA or Europe. In fact, even in USA or Europe, there is only a quite limited number of this kind of chamber simulation. Therefore we think it is necessary to conduct this kind of study in China for diesel vehicles because the engine technology and fuel quality might be quite different. And a large audience, including policy makers, would be interested in the results.

Q: The authors also argue that the higher SOA production observed from their engines compared to Gordon et al and Chirico et al is a significant result. However, this could merely be a result of less dilution of the exhaust in the smog chamber and resultant partitioning during oxidation. Figure 2 shows that the OA concentration in the chamber after exhaust injection was 50 ug/mËĘ3. This means that the chamber had ample POA for SOA to partition into, and oodles of vapor available to oxidize and form SOA. My impression is that previous similar studies worked with much lower POA concentrations, and therefore produced less SOA. The authors have done nothing to convince me that the excess SOA formed in their experiments is the result of higher SOA formation potential of the exhaust rather than higher initial concentrations.

Reply: The dilution ratio was 66-214 (see Table 2 in the revised version) comparable with 160 (Gordon et al., 2014b), and 59-94 for no aftertreatment experiments (Chirico et al., 2010). We agree that higher SOA could be caused by higher POA. What is important that our results revealed lower ratios of SOA/POA. As discussed the in revised manuscript, POA in our study had much higher O/C ratio and less volatility. This can

partly explain higher initial concentrations and less secondary formation (much more oxidized).

Q: The results presented in the manuscript show that a significant amount of SOA is formed during photo-oxidation of dilute exhaust, and that the OA becomes more oxidized during oxidation. This is not new news - Sage et al showed this exact behavior in ACP in 2008 (albeit with a laboratory-scale diesel engine), and the result has been repeated multiple times with emissions from many sources (gasoline engines, diesel engines, aircraft, etc). While I think the underlying data are sound, and the methods appropriate, simply regurgitating previously-published experiments should not pass muster for publication in ACP. The authors need to show how their study adds substantially to the existing state of knowledge.

Reply: In this study, we investigated the possible SOA precursors, and found that significant glyoxal and methylglyoxal formed during the photochemical aging. We noticed that compared to previous studies, the SOA/POA ratios were lower, which could be caused by the difference in the volatility of emitted organics. To a certain extent, these findings help better understand the atmospheric evolution of diesel exhaust and SOA formation in China. Additionally, whether gasoline or diesel dominates the vehicular OA is still a controversial issue (Gentner et al., 2012; Jathar et al., 2014). In this paper, we estimated the gasoline derived OA and diesel derived OA, and found that diesel derived OA dominated over gasoline derived OA in China. We also observed the degradation of particle-phase semi-volatiles like PAHs.

Specific comments: - The manuscript needs a thorough English grammar check.

Reply: The language of the revised manuscript has been edited by a native English speaker.

- Is Figure 1 comparing apples to apples? E.g., does it compare the w=0 case from the present study to w=0 from previous work (or w=1 to w=1)?

Reply: The EFs/PFs in the present study and in previous works were all compared at $\omega=1$ case.

- Line 330-333: Increase in m/z 59 during oxidation could also be acetone

Reply: Acetone is mainly biogenic (Jacob et al., 2002), or produced via oxidation of C3-C5 isoalkanes (Fu et al., 2008). We measured the VOCs offline and list in Table 1. During the chamber experiments, the mixing ratios of C3-C5 alkane were only 1-2 ppbv, therefore, the signal of m/z 59 is mainly glyoxal. (Lines 347-350 in the revised manuscript).

- Figure 4 - which line goes with which axis?

Reply: Revised as suggested.

- The authors only seem to use experiment 8 for representative data. This makes me wary about results from the other experiments.

Reply: Revised as suggested, more data from other experiments were presented in the revised manuscript.

- Figure 8a - what does SOA mean? Is this the total OA at the end of the experiment or was SOA separated from POA somehow?

Reply: SOA is the aged OA at the end of the experiment. It has been corrected to "aged OA" in revised manuscript.

- Fig 8b would be better if it showed VK plots for more than one experiment.

Reply: Revised as suggested.

Please also note the supplement to this comment:
http://www.atmos-chem-phys-discuss.net/acp-2016-50/acp-2016-50-AC2-supplement.zip

---

## Author Comment (AC3) · 26 Apr 2016

Q: Deng et al. present results showing that diesel exhaust injected into a smog chamber produces SOA when the lights are switched on. This result is far from unexpected, as previous studies have shown the same. Reading through the article gives the impression that the authors followed a recipe. If anything, the main contribution of this work would be to demonstrate that a diesel car without after treatment is the same in China as anywhere else. Thus, considering ACP's scope '[ACP] is focused on studies with general implications for atmospheric science rather than investigations that are primarily of local or technical interest', I feel that this articles falls under the latter category, particularly with regard to local interest. But perhaps the authors can add

something. Therefore, aside from criticism of the article's scope,

Reply: As diesel vehicle exhaust is an important source of air pollutants (including BC) with regional impacts, primary emission of diesel exhaust in China and their secondary products formed during atmospheric aging is an issue of wide concern. Quite opposite to the situation in Europe, in China about 80% of motor vehicles are gasoline-driven. If we change more gasoline cars to diesel ones in the future for the control of surface ozone pollutions in the future, we need to know if SOA formation from diesel vehicle exhaust is the same as that in USA or Europe. This kind of study is necessary in China as the engine technology and fuel quality are quite different. We found SOA/POA ratios much lower than previous studies. We estimated that diesel derived OA dominated over gasoline derived OA in China although number of gasoline vehicles are about 4 times that of diesel vehicles. We noticed that our POA had a higher O/C ratio with less volatility. We also tried to investigate more SOA precursors in the gas phase (like glyoxal and methylglyoxal) and in the particulate phase (like the PAHs).

there are other major issues of concern: Q: The grammar needs improvement throughout. While perfection does not need to be the goal, the language should not get in the way of understanding the research.

Reply: The language of the revised manuscript has been edited by a English native speaker.

Q: I also have a concern regarding data represented in Fig. 2C. The authors use the black carbon (BC) time series to quantify organic aerosol (OA) wall losses. Firstly, looking at the first hour after lights on, BC does not decrease. Despite this, wall lost OA is added (since the raw OA trace is decreasing after around 30 minutes). Secondly, variation in the raw trace does not match variation in the wall loss corrected traces. See for example the small perturbation just after t=-1h, not represented in the wall loss corrected trace. Thirdly, variation in the BC trace is not present in the OA trace for the w=1 case.

Reply: As shown in Figure 2C, after the lights were turned on, the measured OA mass increased, demonstrating significant SOA formed. Moreover, as described in manuscript, the newly formed SOA was coated on preexisting particles; this could enhance the BC mass absorption efficiency and artificially increase the estimated BC concentration (Schnaiter et al., 2005; Shiraiwa et al., 2010). Therefore, BC didn't decrease for first hour after lights on. To eliminate this effect, we corrected the wall loss by using exponential fit to the BC data (equation 4, now is equation 5). Therefore, the variation in BC trace is not presented in the OA trace for the $\omega$=1 case. The data of wall loss corrected OA was 10 min averaged to smooth the curve, therefore, the small perturbation not represented in the wall loss corrected trace. That has been corrected in the revised manuscript.

Q: I also note that Equation 2 (w=0 wall loss) does not give the time dependent suspended OA mass, while equation 3 (w=1) does. Can the authors please provide here an equivalent expression for suspended OA mass as a function of time (i.e. the equation used to calculate corrected OA mass), and give assurance that the data, as calculated, is that which is shown?

Reply: Thank you. We have added an equation (Equation 3 in the revised version) to calculate corrected OA mass: ãĂŰOAãĂŮ_(total,t)= ãĂŰOAãĂŮ_sus (t)+ ∫ _0^t$k$·ãĂŰOAãĂŮ_sus (t)dt (3)ãĂŮ

Please also note the supplement to this comment:
http://www.atmos-chem-phys-discuss.net/acp-2016-50/acp-2016-50-AC3-supplement.zip